# Latent Boost: Leveraging Latent Space Distance Metrics to Augment Classification Performance

## Abstract

The pursuit of boosting classification performance in Machine Learning has primarily focused on refining model architectures and hyperparameters through probabilistic loss optimization. However, such an approach often neglects the profound, untapped potential embedded in internal structural information, which can significantly elevate the training process. In this work, we introduce *Latent Boost*, a novel approach that incorporates the very definition of classification via latent representation distance metrics to enhance the conventional dataset-oriented classification training. Thus during training, the model is not only optimized for classification metrics of the discrete data points but also adheres to the rule that the collective representation zones of each class should be sharply clustered. By leveraging the rich structural insights of high-dimensional latent representations, *Latent Boost* not only improves classification metrics like F1-Scores but also brings additional benefits of improved interpretability with higher silhouette scores and steady-fast convergence with fewer training epochs. *Latent Boost* brings these performance and latent structural benefits with minimum additional cost and no data-specific requirements.

## 1 Introduction

Traditional data-driven classification training often employs a black-box approach that focuses solely on optimizing classification scores for discrete datasets, neglecting important aspects such as clusters within the continuous representation. In contrast, we explore an innovative approach by explicitly integrating the classification task into the latent representation through distance metric learning. In this work, we introduce a novel method that seamlessly integrates latent cluster distance metrics into probabilistic training, fundamentally transforming the paradigm of structured latent representations. While conventional probabilistic approaches typically center on individual data samples, they often overlook the intricate relationships among data points, particularly within clusters, where the interdependencies and structural nuances are essential for capturing the underlying information. In other words, while the end-to-end classification performance might be satisfactory, the results like the F1-Score are derived from discrete data points. Internally, the collective clusters of different classes could still be cluttered in the continuous latent representation, as the training processes only optimize for the probabilistic loss of the dataset. In contrast, our innovative approach guarantees that semantically similar data points are thoroughly aligned in proximity, while dissimilar points are distinctly separated, thereby enhancing the model's capacity for nuanced differentiation.

Although distance metric learning has established its efficacy in Machine Learning—particularly in clustering and pre-training tasks (Kulis et al., 2013; Kaya & Bilge, 2019)—its application in classification has been limited to simpler models like K-nearest Neighbors (Cover & Hart, 1967) and Support Vector Machines (Cortes, 1995). We assert that intermediate latent representations hold critical structural information essential for class distributions, and enhancing their cluster separation can significantly boost classification performance. To the best of our knowledge, *Latent Boost* is the first method to integrate distance metrics into the classification loss function, empowering the model to cultivate more meaningful and discriminative features without requiring explicit supervision.

Our key contributions include:

- We propose a distance-based loss *Latent Boost* which is inspired by the Magnet loss, addressing previously overlooked nuisances with dynamic adaptation and discriminative information density.

- We identify the advantages of fusing probabilistic loss distance-based metrics extracted from latent layers through a weighted sum loss equation.

- We demonstrate *Latent Boost* is a simple yet efficient approach to enhance performance and interpretability while shrinking computational demand through faster convergence.

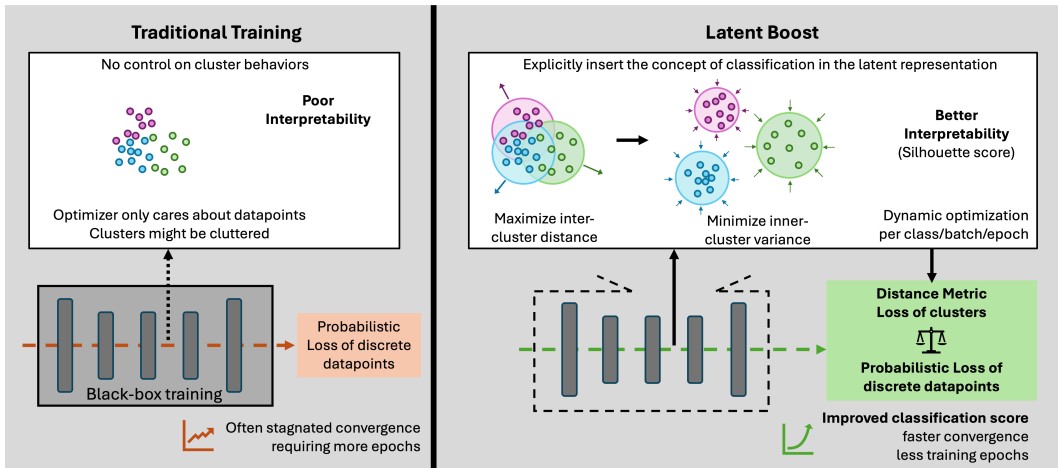

Figure 1: Oppose to traditional training, relying on probabilistic loss only, *Latent Boost* injects distance metric information, obtained from the model's hidden latent representations, as addition into the training through balanced weighted sum equations.

## 2 RELATED WORK

Distance Metric Learning has emerged as a crucial area in Machine Learning, offering a wide range of techniques aimed at improving performance in tasks of primarily unsupervised clustering and retrieval of latent representation information in general (Kulis et al., 2013; Wang & Sun, 2015). Such loss functions generally focus on minimizing intra-cluster variance and maximizing inter-cluster distances by learning a latent representation where similar points are closer together, and dissimilar points are further apart (Kaya & Bilge, 2019). These functions penalize the model until the latent representation aligns with the desired distance metric priorities.

The structure and information density in latent representations also improves interpretability. Discriminative Dimension Selection can be utilized to enhance the interpretability and clustering performance, especially for K-means clustering by selectively retaining relevant features as proposed by Lian et al. (2024). Similarly, works on adaptive feature selection have focused on developing efficient strategies to reduce the burden of complex dimensionality, such as Zhou & He (2024), improving clustering accuracy across several benchmarks. Oppose to utilizing the Euclidean distance information, a boosting algorithm to effectively learn Mahalanobis distance metrics was proposed by Chang (2012), demonstrating its effectiveness on popular datasets to capture intrinsic distance relationships. Mahalanobis-based techniques have been widely adopted, with approaches such as Large Margin Nearest Neighbor (LMNN) (Weinberger & Saul, 2009) maximizing the margin between different classes, and Information-Theoretic Metric Learning (ITML) (Davis et al., 2007), which minimizes the relative entropy between distance distributions.

Pairwise and Contrastive loss functions have shown remarkable improvements in distance metric learning, such as the Contrastive loss function introduced by Hadsell et al. (2006) for learning dimensionality-reducing embeddings. This principle has been adopted in domains like zero-shot

learning (Wang & Chen, 2017), cross-modal retrieval (Wang et al., 2017), and large-scale face recognition (Liu et al., 2017). Contrastive learning has also been used in self-supervised settings, where methods such as SimCLR (Chen et al., 2020) leverage this loss to learn robust features without labels. The promising approaches of triplet-based methods for distance metric learning have been challenging to optimize due to the need for finding informative triplet anchor points (Do et al., 2019). To address this, semi-hard triplet mining methods like Schroff et al. (2015) have been developed, leading to more efficient training. Advanced sampling strategies have been proposed to improve the performance of triplet-based learning systems Hermans et al. (2017). Additionally, using proxy points to approximate original data points, as shown by Movshovitz-Attias et al. (2017), further improves convergence and stability in Triplet loss optimization. Magnet loss has gained attention in recent years as an effective method for distance metric learning, particularly in dealing with high-dimensional data where traditional losses like Triplet loss face challenges (Rippel et al., 2015). Yu et al. (2020) introduces Semantic Drift Compensation as part of Magnet loss as a method to address catastrophic forgetting in class-incremental learning by estimating and compensating for the feature drift of previous tasks, significantly improving performance in embedding networks without requiring exemplars. By focusing on the distribution of latent representations within each class, Magnet loss works to minimize the overlap between class clusters, thus providing better discrimination, especially in face identification in noisy and large-scale datasets (Deng et al., 2020).

Metric learning with constraints has led to significant advances, with pairwise constraints being integrated into methods like Constrained Clustering via Metric Learning (Bilenko & Basu, 2004). Similarly, Ding & Li (2007) extended this concept into semi-supervised clustering, demonstrating the utility of leveraging partial label information. In sparse and high-dimensional data scenarios, sparse subspace clustering (Liu et al., 2010) has been a successful approach.

Methods like DeepCluster (Caron et al., 2018) and the cluster-based contrastive learning proposed by Caron et al. (2020) demonstrate how metric learning can generate meaningful representations for downstream tasks such as image retrieval. The innovation in these approaches lies in integrating clustering with deep learning techniques to build robust data representations. Hybrid models that integrate different learning techniques have also gained attention. For example, Lee et al. (2018) employed stacked attention networks for cross-modal tasks, combining textual and visual data within joint latent spaces to enable multi-modal learning.

Fairness in distance metric learning has also become a crucial area of focus, with works like Lahoti et al. (2020) exploring how adversarial techniques can be used to ensure fairness in learned distance metrics. These methods prevent demographic bias and ensure equitable performance across different groups, a significant consideration for real-world applications.

Based on the previously referenced works, we identified the Contrastive, Triplet, N-pair, and Magnet loss as the currently most relevant and recent choices wherefore we outline their concepts in the following:

**Contrastive Loss** is one of the simplest and most widely used loss functions for distance metric learning, introduced in the context of training Siamese networks (Chopra et al., 2005). The goal of Contrastive loss is to minimize the distance between pairs of samples that are similar and maximize the distance between pairs of samples that are dissimilar up to a certain margin. The Contrastive loss function is formulated in Equation (1), where $y_i \in \{0, 1\}$ is a binary indicator of similarity, with $y_i = 1$ for similar pairs and $y_i = 0$ for dissimilar ones. The Euclidean distance $d_i$ is calculated between the latent representations of the two samples. The equation is applied across the total number of pairs $N$. Additionally, $m$ works as a penalization, defining the margin as the minimum distance for dissimilar pairs.

$$\mathcal{L}_{\text{contrast}} = \frac{1}{2N} \sum_{i=1}^{N} \left( y_i \cdot d_i^2 + (1 - y_i) \cdot \max(0, m - d_i)^2 \right) \tag{1}$$

**Triplet Loss** was popularized by the FaceNet model architecture, introducing the triplets of samples (Schroff et al., 2015) as shown in Equation (2). An anchor $a_i$, a positive sample $p_i$, and a negative sample $n_i$ form one triplet. The goal is to ensure that the distance between the anchor and the positive sample is smaller than the distance between the anchor and the negative sample by at least a margin $\alpha$. Compared to Contrastive loss, the Triplet loss is based on the anchor as opposed to calculating it solely on the pairwise samples. The Euclidean distance is calculated between the representations of

the positive and negative sample to its anchor with $m$ as the margin term.

$$\mathcal{L}_{\text{triplet}} = \frac{1}{N} \sum_{i=1}^{N} \max\left(0, d(a_i, p_i) - d(a_i, n_i) + m\right) \tag{2}$$

**N-pair Loss** shares similarity with Triplet loss and aims to converge more stable across the clusters (Sohn, 2016). Instead of using a single negative sample per triplet, N-pair loss optimizes the distance between the anchor and the positive sample while contrasting it with multiple negatives simultaneously. N-pair loss is particularly effective in multi-class classification tasks for large-scale datasets. The N-pair loss is defined in Equation (3) ($\mathbf{f}_i$ is the embedding of the anchor sample, $\mathbf{f}_i^+$ and $\mathbf{f}_i^-$ are positive and negative samples)

$$\mathcal{L}_{\text{N-pair}} = \frac{1}{N} \sum_{i=1}^{N} \log\left(1 + \sum_{i^+ \neq i^-} e^{\mathbf{f}_i^T \mathbf{f}_i^- - \mathbf{f}_i^T \mathbf{f}_i^+}\right) \tag{3}$$

**Magnet Loss**, introduced by Rippel et al. (2015), is designed to address the challenges of high-dimensional and complex data distributions by grouping data into clusters instead of focusing on pairwise or triplet calculations. To benefit from cluster-based information, Magnet loss penalizes a sample based on its distance to the centroid of the correct cluster and the other false clusters. The behavior of a Magnet is obtained by accounting for both intra-class compactness and inter-class separation. The Magnet loss function is formulated in Equation (4), where $r_n$ is the latent representation of the sample, $\mu(r_n)$ is the mean of the cluster containing $r_n$, and $\mu_c$ are the means of other clusters. $\sigma$ is the standard deviation, $\alpha$ is a margin term. $N$ is the total number of samples whereas $K$ represents the number of clusters.

$$\mathcal{L}_{\text{magnet}} = \frac{1}{N} \sum_{n=1}^{N} \left\{ -\log\left( \frac{e^{-\frac{1}{2\sigma^2} \|\mathbf{r}_n - \boldsymbol{\mu}(r_n)\|_2^2 - \alpha}}{\sum_{c \neq C(r_n)} \sum_{k=1}^{K} e^{-\frac{1}{2\sigma^2} \|\mathbf{r}_n - \boldsymbol{\mu}_k^c\|_2^2}} \right) \right\} \tag{4}$$

## 3    LATENT BOOST METHODOLOGY

Among the hitherto loss functions, Magnet loss provides superior potential in enhancing classification performance as the following section 4.2 experiments. However, no modifications or optimizations of the hyperparameters associated with the Magnet loss have been explored so far. Additionally, in the original formulation, the distance metric loss is computed across the full latent space. While this approach captures the complete available information, in early training epochs, this might introduce noise if the latent representation includes dimensions that do not yet provide beneficial information.

To address this, we propose incorporating a dimensionality reduction step before calculating the *Latent Boost* batch loss by applying Principal Component Analysis (PCA) to reduce the dimensions of the latent vectors. Let $\mathbf{W}_{\text{PCA}} \in \mathbb{R}^{d' \times d}$ denote the matrix of the top principal components, where $d'$ is the reduced dimension and $d$ is the original dimension. Each latent vector $r_n$ is projected onto the lower-dimensional subspace as:

$$r_n' = \mathbf{W}_{\text{PCA}} r_n \tag{5}$$

Similarly, the cluster centroids $\mu_{r_n}$ and $\mu_{c_k}$ are projected onto the same subspace as:

$$\mu_{r_n}' = \mathbf{W}_{\text{PCA}} \mu_{r_n}, \quad \mu_{c_k}' = \mathbf{W}_{\text{PCA}} \mu_{c_k} \tag{6}$$

The number of retained principal components $dim$ is determined by ensuring a predefined threshold $T$ of cumulative explained variance is met, as follows:

$$dim = \min\left\{ i \left| \frac{\sum_{j=1}^{i} \frac{S_j^2}{m-1}}{\sum_{j=1}^{\text{max\_dim}} \frac{S_j^2}{m-1}} \geq T \right. \right\} \quad \text{or} \quad dim = dim_{max} \text{ if no such } i \text{ exists} \tag{7}$$

Here, $S_j$ are the singular values from the Singular Value Decomposition (SVD) that is extracted from the PCA dimension reduction process, $m$ is the number of samples, $n$ is the number of available

features, and $\text{max\_dim} = \min(m, n)$ is the maximum number of possible components. $T$ represents the threshold for cumulative explained variance, which we set to 0.95 based on initial investigations (see Appendix).

After applying PCA, we compute the *Latent Boost* loss using the reduced latent representations $r'_n$ and $\mu'$. The *Latent Boost* loss is based on the orignal Magnet loss from Equation (4), with several enhancements:

First, the variance for each cluster $\sigma_C^2$ is now computed based on the spread of points within each cluster individually. Specifically, the variance represents the average squared distance of the points $r_i$ in cluster $C$ from the cluster mean $\mu_C$. The formula for calculating the variance for a cluster $C$ is given by:

$$\sigma_C^2 = \frac{1}{|C| - 1} \sum_{r_i \in C} \|\mathbf{r}_i - \boldsymbol{\mu}_C\|^2 \tag{8}$$

Here, $|C|$ denotes the number of points in cluster $C$ and adjusts the degree of freedom. $r_i$ represents the position of the $i$-th point in the cluster and $\mu_C$ is the centroid of the cluster from which we calculate the squared Euclidean distance between. This dynamic variance allows the model to adapt to clusters with varying densities, meaning clusters that are more spread out will have larger variance and be more forcefully compressed compared to tighter clusters.

Additionally, we introduce a hyperparameter $\beta$ in the denominator of the loss function. The Magnet loss is composed of two main components: intra-cluster variance minimization, controlled by $\alpha$, and inter-cluster separation, now influenced by $\beta$. To balance these competing objectives, we developed dynamic strategies to adjust $\alpha$ and $\beta$ based on the current training epoch $E$, as formulated in Equation (9). Initially, the focus is on achieving tight clustering of intra-class samples by assigning larger values to $\alpha$. Subsequently, $\beta$ plays a greater role in encouraging separation between clusters. The update rule for $\alpha$ follows an exponential decay schedule due to the simple subtraction as a margin term. It starts at a value of $1 + \alpha_0$, where $\alpha_0$ controls the initial strength, and gradually decreases by the factor $e^{-\frac{E}{1.05 \cdot E_{\text{total}}}}$ towards the commonly set value of 1.0 as training progresses.

In contrast, the update rule for $\beta$ follows a linear schedule, starting from $\beta_0$ and decreases linearly until it reaches zero after the first 20% of the conventional training period. This schedule ensures that $\beta$ gradually diminishes and therefore increases the effect of distancing the clusters. This mechanism allows the model to be unconstrained by $\beta$ in the early stages due to the multiplication factor and its initialization with 1.0, but ensures that the influence of the inter-cluster distance denominator increases for the overall loss.

Based on our experiments, the intra-cluster is mostly relevant during early epochs, whereas the exponential decrease of $\alpha$ helps to diminish the effect of intra-cluster variance distance later in the training progress. However, the inter-cluster importance needs to be linearly increased by decreasing $\beta$ in order to move the clusters continuously and gradually farther apart till the model converges. An exemplary schedule of $\alpha$ and $\beta$ for 100 epochs is shown in Appendix C.

To ensure numerical stability across our experiments, we added $\epsilon$ and set it to $1 \cdot 10^{-8}$ as a small constant to prevent division by zero and underflow of floating point numbers.

$$\alpha = 1 + \alpha_0 \cdot e^{-\frac{E}{1.05 \cdot E_{\text{total}}}} \qquad \beta = \beta_0 \cdot \left(1 - \frac{E}{0.2 \cdot E_{\text{total}}}\right) \tag{9}$$

Finally, the complete *Latent Boost* loss is given by:

$$\mathcal{L}_{\text{LB}} = \frac{1}{N} \sum_{n=1}^{N} \left\{ -\log \left( \frac{e^{-\frac{1}{2\sigma_{C_k}^2} \|\mathbf{r}'_n - \boldsymbol{\mu}'_{r_n}\|_2^2 - \alpha}}{\sum_{c \neq C(r_n)} \sum_{k=1}^{K} e^{-\frac{1}{2\sigma_{C_k}^2} \|\mathbf{r}'_n - \boldsymbol{\mu}'_{c_k}\|_2^2 \cdot \beta}} + \epsilon \right) \right\} \tag{10}$$

Here, $r'_n$ and $\mu'$ represent the reduced-dimensional representations of the samples and cluster centroids, respectively, after applying PCA.

In summary, $\mathcal{L}_{\text{LB}}$ is inspired by Magnet loss with improvements addressing the following aspects, calculated dynamically and independently within every batch:

1. We reduce noisy dimensions using PCA before calculating the loss term.

2. Instead of an indiscriminate variance, we calculate the variance for different class labels, allowing the model to adapt to clusters with varying densities.

3. We developed dynamic update rules to decouple the intra-cluster variance and inter-cluster separation, to first emphasize on shrinking the clusters and then pushing them apart.

# 4 EXPERIMENTS

As another novel contribution, we incorporate distance metric information, which has been hitherto primarily used for unsupervised clustering, into supervised classification with probabilistic loss through a weighted sum loss function. Equation (11) outlines the total loss calculation on which our experiments rest. We introduce the hyperparameter $\lambda$ to balance the weight between the two loss components. The range of $\lambda$ is continuous between 0 to 1, with 0 giving full weight to the probabilistic loss whereas 1 gives full weight to the distance metric loss. To isolate the influence of $\lambda$ and the effect of each selected distance metric, we fixed the probabilistic loss to shrink the overall experiment complexity. We therefore selected the well-established soft-max cross-entropy, which matches the idea of utilizing the weighted sum equation for multi-class classification.

Our preliminary experiments, as evaluated in Section 4.2, explore the standard version of the presented distance metric losses induced with different weights of $\lambda$. On top of that, we replace the distance loss with our custom *Latent Boost* solution (abbreviated $\mathcal{L}_{\text{LB}}$) in Section 3.

$$\mathcal{L}_{total} = \lambda \cdot \mathcal{L}_{\text{dist}} + (1 - \lambda) \cdot \mathcal{L}_{\text{cross-entropy}}$$

$$\text{with } \mathcal{L}_{\text{dist}} \in \{\mathcal{L}_{\text{contrast}}, \mathcal{L}_{\text{triplet}}, \mathcal{L}_{\text{N-pair}}, \mathcal{L}_{\text{magnet}}, \mathcal{L}_{\textbf{LB}}\}, \quad \mathcal{L}_{\text{cross-entropy}} = -\sum_{\forall x} p(x) \log(q(x)) \quad (11)$$

## 4.1 EXPERIMENT SETUP

We selected three different experiment setups of different complexity to explore the effects of the previously introduced distance metric losses through the weighted sum equations. Starting with a simple Convolutional Neural Network (CNN) trained on Fashion-MNIST (Xiao et al., 2017), we trained a VGG-16 model (Simonyan & Zisserman, 2014) on CIFAR10 (Krizhevsky et al., 2009), followed by training a ResNet-50 (He et al., 2016) architecture on CIFAR-100 (Krizhevsky et al., 2009). The full description of each model architecture can be found in Appendix A.1. For our initial investigation in the following section, we kept the distance metric losses hyperparameters constant and initialized to the standard recommendations (see Appendix A.3 for details). For each experiment, we extracted the latent representation after the convolutional part in the network architecture, meaning just before the classification section. For the CNN the latent representation has a dimension of 64, the VGG-16 model obtains 512 dimensions on that layer and the ResNet-50 generates a latent representation of 2048 dimensions. The choice to extract latent features from layers just before the classification head is based on their ability to capture high-level, task-specific representations that balance discriminative power and compactness, as these layers contain semantically rich information essential for classification tasks. Earlier layers typically emphasize low-level features and may lack the necessary information density required for effective downstream processing, making deeper layers more suitable.

Each setup was implemented in PyTorch (Paszke et al., 2019) with a learning rate scheduler that reduced the learning rate by factor 5 when a plateau was reached after 10 epochs. The Adam optimizer was utilized for each experiment to efficiently work out the gradients (Kingma, 2014). We applied early stopping metrics with 20 epochs patience based on the validation accuracy to additionally compare the convergence speed next to the classification performance. Finally, we trained each experiment setup five times to provide meaningful results. We assigned each trial a different random seed, which we kept constant within the trials different runs across the variation of lambda to balance the distance metric with the classic cross-entropy.

## 4.2 PRELIMINARY EVALUATION

In this section, we evaluate the effect of varying the hyperparameter $\lambda$ across different loss functions while keeping the other hyperparameters untouched. The results are reported across our three

Table 1: Baseline comparison of accuracy (↑) and Micro-F1 score (↑) across our three experiments using different loss functions and varying $\lambda$ values; the baseline ($\lambda=0.0$) represents the standard model without additional loss integration.

| Dataset | Loss Type | $\lambda$ | Accuracy (Mean ± Std) | Micro-F1 (Mean ± Std) |
|---|---|---|---|---|
| | Baseline | **0.0** | **88.59 ± 0.15** | **0.8867 ± 0.0020** |
| | | 0.25 | 88.83 ± 0.08 | 0.8884 ± 0.0015 |
| | Contrast | 0.5 | 88.57 ± 0.14 | 0.8850 ± 0.0019 |
| | | 0.75 | 87.79 ± 0.25 | 0.8773 ± 0.0022 |
| | | 0.25 | 89.13 ± 0.18 | 0.8909 ± 0.0011 |
| | Triplet | 0.5 | 89.13 ± 0.34 | 0.8915 ± 0.0036 |
| | | 0.75 | 88.97 ± 0.33 | 0.8900 ± 0.0038 |
| **Fashion MNIST** | | 0.25 | 88.85 ± 0.07 | 0.8883 ± 0.0010 |
| | N-pair | 0.5 | 89.25 ± 0.09 | 0.8923 ± 0.0005 |
| | | 0.75 | 88.71 ± 0.53 | 0.8869 ± 0.0044 |
| | | 0.25 | 89.27 ± 0.29 | 0.8928 ± 0.0023 |
| | Magnet | 0.5 | 89.07 ± 0.21 | 0.8911 ± 0.0022 |
| | | **0.75** | **89.52 ± 0.34** | **0.8946 ± 0.0040** |
| | Baseline | **0.0** | **85.88 ± 0.40** | **0.8586 ± 0.0042** |
| | | 0.25 | 86.01 ± 0.99 | 0.8600 ± 0.0102 |
| | Contrast | 0.5 | 86.87 ± 0.56 | 0.8693 ± 0.0049 |
| | | 0.75 | 84.74 ± 0.44 | 0.8479 ± 0.0045 |
| | | 0.25 | 86.81 ± 0.25 | 0.8667 ± 0.0025 |
| | Triplet | 0.5 | 86.88 ± 0.64 | 0.8683 ± 0.0059 |
| **CIFAR-10** | | 0.75 | 86.95 ± 0.42 | 0.8690 ± 0.0039 |
| | | 0.25 | 84.52 ± 0.71 | 0.8446 ± 0.0067 |
| | N-pair | 0.5 | 86.43 ± 0.78 | 0.8643 ± 0.0081 |
| | | 0.75 | 85.91 ± 0.40 | 0.8583 ± 0.0042 |
| | | 0.25 | 86.91 ± 0.69 | 0.8692 ± 0.0073 |
| | Magnet | 0.5 | 86.72 ± 1.77 | 0.8671 ± 0.0174 |
| | | **0.75** | **87.36 ± 0.82** | **0.8738 ± 0.0081** |
| | Baseline | **0.0** | **61.77 ± 0.49** | **0.6163 ± 0.0064** |
| | | 0.25 | 61.04 ± 0.81 | 0.6088 ± 0.0087 |
| | Contrast | 0.5 | 60.00 ± 0.71 | 0.5984 ± 0.0087 |
| | | 0.75 | 54.91 ± 0.71 | 0.5482 ± 0.0071 |
| | | 0.25 | 62.36 ± 0.91 | 0.6213 ± 0.0070 |
| | Triplet | 0.5 | 60.41 ± 0.51 | 0.6022 ± 0.0041 |
| **CIFAR-100** | | 0.75 | 56.13 ± 0.76 | 0.5597 ± 0.0082 |
| | | 0.25 | 61.54 ± 0.83 | 0.6139 ± 0.0075 |
| | N-pair | 0.5 | 60.13 ± 0.94 | 0.5991 ± 0.0075 |
| | | 0.75 | 47.73 ± 1.79 | 0.4694 ± 0.0168 |
| | | 0.25 | 62.43 ± 0.38 | 0.6242 ± 0.0039 |
| | Magnet | **0.5** | **62.75 ± 0.51** | **0.6256 ± 0.0057** |
| | | 0.75 | 62.66 ± 0.54 | 0.6242 ± 0.0045 |

experiment setups, comparing accuracy and Micro-F1 Scores. The explanation of both metrics can be found in Appendix A.2. Table 1 shows the most relevant $\lambda$ values to compare and recognize trends, whereas more detailed tables with smaller $\lambda$ step size can be found in Appendix B. The results show that adjusting $\lambda$ can improve performance, Magnet loss. On Fashion MNIST, Magnet loss outperforms other methods, achieving 89.52% accuracy and 0.8946 Micro-F1 scores at $\lambda = 0.75$. On CIFAR-10, Magnet loss also achieves the highest results, with an accuracy of 87.36% and a Micro-F1 score of 0.8738 at $\lambda = 0.75$. However, on CIFAR-100, the results were less conclusive due to its high complexity, but Magnet loss still showed some improvement over the baseline. As a result, the Magnet loss is the most robust method across our experiments, especially when selecting $\lambda$ values in the range 0.5 to 0.9, since it properly improves classification performance across datasets by leveraging latent information and reducing intra-class variability with proper priority.

### 4.3 LATENT BOOST'S PERFORMANCE

We conducted the same experiment, based on our three datasets and model combinations, with the adapted *Latent Boost* distance metric. As shown in Section 4.3, we track the accuracy, Micro-F1 Score, and the duration of training epochs. We calculated the average and standard deviation across

our five experiment trials. The selected $\lambda$ values mimic the range of the preliminary experiment. The full results can be found in Appendix D. *Latent Boost* proves to consistently outperform the baseline and the classic Magnet loss results from the previous experiments of Table 1. For Fashion-MNIST and CIFAR-10, $\lambda$ selection of 0.75 and 0.5 for the CIFAR-100 obtained the best performance equal to the original Magnet loss results. The Fashion MNIST shows a 2.56% increase in accuracy and a 1.93% increase in F1 Score, while CIFAR-10 and CIFAR-100 exhibit comparable improvements of around 2-3%. These gains are accompanied by reduced standard deviations, indicating a more stable and reliable convergence due to the additional structural information from latent representation. The tighter variability in performance indicates that *Latent Boost* is more consistent across datasets.

Another advantage of Latent Boost is its faster and more stable convergence next to the model performance. The number of epochs required for Fashion MNIST training is reduced by around 14%, while CIFAR-10 and CIFAR-100 show reductions of around 13% and 21%, respectively. This reduction not only speeds up training but also offers potential sustainability benefits by decreasing computational resources and energy consumption.

As stated before, the threshold for selecting the number of principal components was decided to be at 95% of cumulative explained variance. For Fashion-MNIST, the number of principal components ranged around 45 to 50, with slight degradation to 40 components when the latent representation reaches a sufficient structure. For CIFAR-10, the number of principal components started around 65 and decreased to 40 components, stating that the latent representation is well structured throughout the epochs. For the final CIFAR-100 experiment, the principal components used were around 100 to 110 without a decent decrease over time. That being said, it reflects our findings of struggling to structure the large latent representation properly.

Table 2: Accuracy ($\uparrow$), Micro-F1 Score ($\uparrow$), and epoch duration ($\downarrow$) between baseline ($\lambda = 0$), standard Magnet loss and the *Latent Boost* on the unseen test dataset (best $\lambda$ selection); percentage improvement compares *Latent Boost* with the baseline for each metric.

|  | Metric | Baseline | Magnet | Latent Boost | Improv. (%) |
|---|---|---|---|---|---|
| **Fashion MNIST** | Accuracy | 88.59 ± 0.15 | 89.52 ± 0.34 | **90.86 ± 0.21** | 2.56% |
|  | Micro-F1 | 0.8867 ± 0.002 | 0.8946 ± 0.004 | **0.9038 ± 0.005** | 1.93% |
|  | Nr. Epochs | 40.67 ± 4.19 | 37.67 ± 6.18 | **34.67 ± 3.09** | -14.76% |
| **CIFAR-10** | Accuracy | 85.88 ± 0.40 | 87.36 ± 0.82 | **88.44 ± 0.28** | 2.98% |
|  | Micro-F1 | 0.8586 ± 0.0042 | 0.8738 ± 0.0081 | **0.8843 ± 0.0024** | 2.99% |
|  | Nr. Epochs | 76.33 ± 8.65 | 72.67 ± 1.70 | **66.33 ± 4.50** | -13.09% |
| **CIFAR-100** | Accuracy | 61.77 ± 0.49 | 62.75 ± 0.51 | **63.04 ± 0.48** | 2.06% |
|  | Micro-F1 | 0.6163 ± 0.0064 | 0.6256 ± 0.0057 | **0.6288 ± 0.0051** | 2.03% |
|  | Nr. Epochs | 86.67 ± 9.46 | 69.67 ± 8.73 | **68.67 ± 6.02** | -20.74% |

## 4.4 LATENT REPRESENTATION

Due to the increase in performance as stated in the previous section, we additionally want to evaluate the structure of the latent representation. Even though we only utilize the distance information from the model's latent representation to calculate the metric and not the data itself, the latent representation within the model should include some enhancements in order to reflect the performance.

In Figure 2, we plotted the 2-dimensional representation of the high-dimensional latent space for each of our experiments. We passed the test dataset of each experiment through the model and extracted the latent representation the same way as previously proposed. However, we do not calculate the *Latent Boost* metric but rather utilize the data and shrink its dimension with classic and state-of-the-art TSNE (Van der Maaten & Hinton, 2008) for visualization purposes. We utilized the TSNE according to best practices, keeping hyperparameters in its standard initialization and fixing the random seed. The visualizations from left to right show the latent representation for the best trial of baseline training without any distance metric information, the Magnet loss training from our preliminary experiments, and the *Latent Boost* approach. From top to bottom, we show each of the experiment models and dataset combinations.

Starting with the Fashion MNIST, even though the clusters were separated properly in the baseline already, we can see that in the *Latent Boost* clusters are formed with more concentrated density and classes such as *brown* and *grey* are well separated. In the scenario with CIFAR-10, a similar trend can

be recognized. Even though there is much more confusion within the latent representation compared to the Fashion MNIST, clusters are more dense in the final stage of *Latent Boost*. Additionally, the main confusion between *green* and *red* class from the baseline could partially be resolved. Some classes are much better formed for clustering and boundaries between the classes can be recognized stronger. For the final experiment on CIFAR-100, the visualization is not meaningful enough to show a clear separation of clusters. This however may be the issue of visualizing the 100 classes with different color shades. Only the surrounding areas of the latent representation form small clusters without significant visible impact.

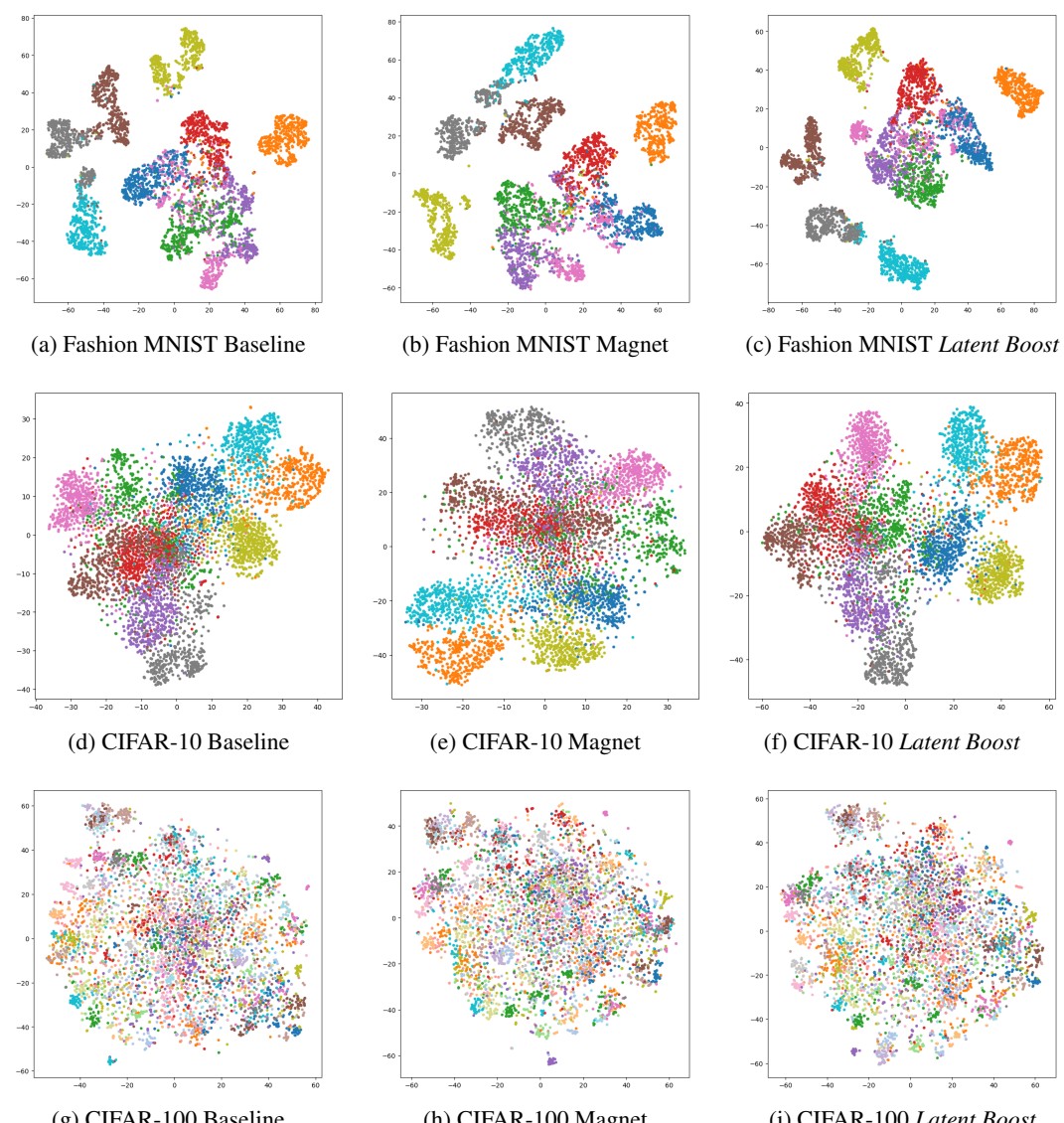

(a) Fashion MNIST Baseline     (b) Fashion MNIST Magnet     (c) Fashion MNIST *Latent Boost*

(d) CIFAR-10 Baseline     (e) CIFAR-10 Magnet     (f) CIFAR-10 *Latent Boost*

(g) CIFAR-100 Baseline     (h) CIFAR-100 Magnet     (i) CIFAR-100 *Latent Boost*

Figure 2: Comparison of baseline ($\lambda$=0), standard Magnet loss, and our *Latent Boost* approach across the three experiment setups.

Since TSNE compresses the latent space and therefore loses the detailed information from the full dimensional representation, the visualizations can only be utilized for visual cross-comparison but do suffice for compelling quantitative evaluation. To quantify the density of clusters and their separation from each other in the original dimension, we selected the Silhouette Score to measure the quality of the latent representation. Originally proposed by Rousseeuw (1987), the Silhouette Score is calculated for each data point by comparing the cohesion within its own cluster and the separation from the nearest neighboring cluster. It has been widely used for quantifying representation inter-

pretability in recent works (Bagirov et al., 2023; Januzaj et al., 2023; Du et al., 2024). The score ranges from $-1$ to 1, with -1 indicating the location or assignment of samples to the wrong cluster, whereas 0 is between clusters and 1 is the best-case with clear allocation to the correct cluster. For a given data point $i$ in a cluster $C_i$, the Silhouette Score $s(i)$ is calculated following Equation (12). $a(i)$ represents the cohesion as to how closely related a data point is to its own cluster, whereas $b(i)$ represents the separation, meaning the distance between a data point to its nearest but not own neighbor cluster.

$$\text{Silhouette Score} = \frac{1}{N} \sum_{i=1}^{N} \frac{b(i) - a(i)}{\max(a(i), b(i))}$$

$$a(i) = \frac{1}{|C_i| - 1} \sum_{j \in C_i, j \neq i} d(i,j) \qquad b(i) = \min_{C \neq C_i} \left( \frac{1}{|C|} \sum_{j \in C} d(i,j) \right)$$

(12)

To evaluate the impact of different training approaches on the latent representation, we calculated the Silhouette Scores for the previously discussed latent representations in Figure 2. The results, shown in Table 3, highlight that both the Magnet loss and *Latent Boost* methods improve cluster separation on the full dimension, relative to the baseline. Latent Boost yields the highest score in all cases, with particularly strong improvements observed for CIFAR-10. The results reflect the preliminary experiment hypothesis across the three datasets. Checking the improvement between baseline and *Latent Boost*, the Fashion MNIST experiment showed slight improvement, since the baseline latent space was already well separated. The CIFAR-10 achieved greatest improvement in the dimensional representation with around 168 %, restating the model's complexity potential for improvement, especially in the high dimensions. For the CIFAR-100, even though some minor improvement could be recognized, the *Latent Boost* approach does not sufficiently support the training as expected. Since the Silhouette Score ranges in negative values for CIFAR-100, major confusion between the large set of classes still dominates the classification despite the slight increase.

Table 3: Silhouette score to quantify cluster separation (larger values in range -1 to 1 represent greater separation); calculated on the models' latent representation without compression from the unseen test dataset; the percentage improvement compares *Latent Boost* with the baseline.

| | **Baseline** | **Magnet** | **Latent Boost** | **Improvement** (%) |
|---|---|---|---|---|
| **Fashion MNIST** | 0.347 | 0.375 | **0.395** | 13.84% |
| **CIFAR-10** | 0.131 | 0.186 | **0.351** | 167.94% |
| **CIFAR-100** | -0.280 | -0.264 | **-0.250** | 10.71% |

While the Silhouette Score captures the quality of latent representation, it is important to note that the model is not directly trained on the latent representation data. Instead, the calculated metrics are injected as additional information through the loss function, helping the model to indirectly improve its internal representation structures. This approach leverages the power of the latent representation without requiring explicit supervision, allowing the model to develop more meaningful features.

## 5 CONCLUSION

Overall, *Latent Boost* explicitly inserts the very definition of classification in the latent layer of an otherwise black-box model, which stimulates the model to produce better-separated clusters in the target latent layer while optimizing for the classification task. This consequentially leads to improved classification results, enhanced interpretability metrics, and more efficient training convergence. While the classification metrics uplift is moderate, the improvement is consistent across all benchmarks with the least deviation. By leveraging both intra-class compactness and inter-class separation in a dynamic manner, *Latent Boost* proves its ability to adapt latent representations for more robust feature learning. One limitation of our approach requires a few epochs to take effect to gradually manipulate the clusters, thus the benefits are diminished in cases with already subnormal training epochs. Furthermore, *Latent Boost* relies on the distance to the cluster center, which assumes a high-dimensional sphere in the latent representation. It remains to be investigated whether other cluster formation strategies can further enhance the performance of the method in question.

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

## A    EXPERIMENT SETUP DETAILS

### A.1    MODEL ARCHITECTURES

The model architecture trained on Fashion MNIST (Xiao et al., 2017) is a convolutional neural network (CNN) adapted for grayscale images. The network starts with a convolutional layer that takes single-channel (grayscale) 28x28 images and outputs a set of feature maps, followed by ReLU activation and max-pooling. The model employs two convolutional layers, with each followed by pooling and dropout of 25% to mitigate overfitting. The resulting feature maps are flattened and passed through fully connected layers. From the flattening layer, we extract the latent features. The final fully connected layer outputs the class probabilities for the 10 fashion categories.

The model architecture used for CIFAR-10 (Krizhevsky et al., 2009) is based on the VGG-16 network (Simonyan & Zisserman, 2014). VGG-16 is a deep convolutional network known for its success in image classification tasks on complex color images like CIFAR-10. This architecture consists of 16 layers, with 13 convolutional layers interspersed with ReLU activations and max pooling to downsample the feature maps, followed by three fully connected layers. The network extracts progressively richer features from the images, which are then classified into one of the 10 classes.

The model trained on CIFAR-100 (Krizhevsky et al., 2009) is based on the ResNet-50 architecture (He et al., 2016) and comprises several key layers to efficiently process the input images. The architecture begins with a modified convolutional layer that accepts three input channels and employs a kernel size of 3 times 3, followed by batch normalization and ReLU activation. It includes four residual blocks (layer1 to layer4), each containing a series of convolutional layers, batch normalization, and skip connections to enhance gradient flow. The network concludes with an average pooling layer, which is flattened to extract the latent representations and a fully connected layer that outputs the final class predictions, specifically tailored for the CIFAR-100 classification task.

### A.2    EVALUATION METRICS

Across our work, we utilized the accuracy and the Micro-F1 Score as the main metrics to validate our approaches and experiments. We selected the Micro-F1 Score to aggregate the contributions of all classes to the overall results.

The Micro-F1 score is given in the range 0 to 1 and defined as:

$$\text{Micro-F1} = 2 \cdot \frac{\text{Micro Precision} \cdot \text{Micro Recall}}{\text{Micro Precision} + \text{Micro Recall}}$$

$$\text{with Micro Precision} = \frac{TP_{\text{total}}}{TP_{\text{total}} + FP_{\text{total}}}, \quad \text{Micro Recall} = \frac{TP_{\text{total}}}{TP_{\text{total}} + FN_{\text{total}}} \tag{13}$$

The Accuracy is given in percentage values and defined as follows:

$$\text{Accuracy} = \frac{TP + TN}{TP + TN + FP + FN} \tag{14}$$

For both, the following abbreviations apply:

- $TP$ = True Positives
- $TN$ = True Negatives
- $FP$ = False Positives
- $FN$ = False Negatives

### A.3    DEFAULT DISTANCE METRIC HYPERPARAMETERS

Table 4 summarizes the original hyperparameter settings for various distance metric loss functions used in our preliminary experiments. For Contrastive loss, the positive and negative margins are set to 0.0 and 1.0, respectively, which determine the threshold for distinguishing between positive

and negative pairs. The Triplet loss employs a margin of 0.05 and considers all possible triplets per anchor, ensuring a comprehensive evaluation of relative distances in the latent representations. N-Pair loss utilizes a MeanReducer as its reducer, which averages the distances, while Magnet loss is configured with an $\alpha$ value of 1.0, controlling the aggressiveness for forcing the latent space separation. These initial values were chosen based on established practices in the literature to ensure a fair comparison independently of the distance metric initialization or adaptation.

| Loss Function | Hyperparameter | Initial Value |
|---|---|---|
| Contrast | positive margin | 0.0 |
| | negative margin | 1.0 |
| Triplet | margin | 0.05 |
| | triplets per anchor | all |
| N-Pair | reducer | MeanReducer |
| Magnet | $\alpha$ | 1.0 |

Table 4: Original hyperparameter setting for each distance metric utilized in the preliminary experiment.

## B   LAMBDA VARIATION IN PRELIMINARY EXPERIMENT

The Tables 5 to 7 present the complete set of $\lambda$ variations in the range of 0.1 to 0.9 across our weighted sum equation to combine distance and probabilistic loss. The results cover the four selected experiments of traditional distance metrics trained on the three setups of models and datasets.

## C   LATENT BOOST HYPERPARAMETER

As an addition to the introduced scheduling of $\alpha$ and $\beta$ within our *Latent Boost* approach as introduced in Section 3, we present an exemplary schedule of hyperparameters across 100 epochs in Figure 3. Throughout our experiments, we initialized and dynamically changed the two hyperparameters according to that process.

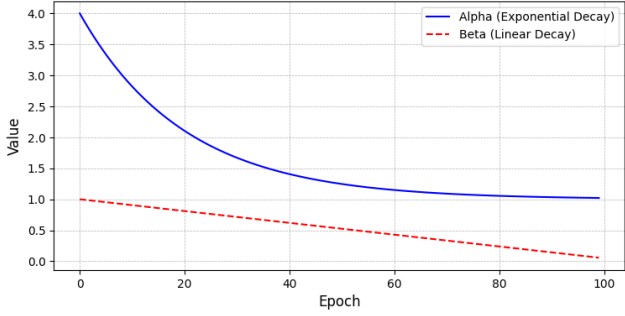

Figure 3: Example based on 100 epochs for the exponential decrease of $\alpha$ and linear decrease of $\beta$ to dynamically adapt the importance of intra- and inter-class loss between cluster in the *Latent Boost* approach.

## D   LATENT BOOST LAMBDA VARIATIONS

The following table outlines the complete set of $\lambda$ variations in range of 0.1 to 0.9 for the *Latent Boost* approach, also covering the unpromising selection of $\lambda$

Table 5: Baseline comparison with extended $\lambda$ values for Fashion MNIST

| Dataset | Loss Type | $\lambda$ | Accuracy (Mean ± Std) | Micro-F1 (Mean ± Std) |
|---|---|---|---|---|
| | Baseline | 0.0 | 88.59 ± 0.15 | 0.8867 ± 0.0020 |
| | | 0.1 | 88.71 ± 0.10 | 0.8870 ± 0.0017 |
| | | 0.2 | 88.78 ± 0.12 | 0.8875 ± 0.0016 |
| | | 0.25 | 88.83 ± 0.08 | 0.8884 ± 0.0015 |
| | | 0.3 | 88.85 ± 0.12 | 0.8887 ± 0.0018 |
| | Contrast | 0.4 | 88.90 ± 0.11 | 0.8893 ± 0.0017 |
| | | 0.5 | 88.57 ± 0.14 | 0.8850 ± 0.0019 |
| | | 0.6 | 88.45 ± 0.15 | 0.8842 ± 0.0020 |
| | | 0.7 | 88.33 ± 0.16 | 0.8830 ± 0.0021 |
| | | 0.75 | 87.79 ± 0.25 | 0.8773 ± 0.0022 |
| | | 0.8 | 87.55 ± 0.22 | 0.8749 ± 0.0020 |
| | | 0.9 | 87.30 ± 0.25 | 0.8725 ± 0.0025 |
| | | 0.1 | 88.95 ± 0.18 | 0.8890 ± 0.0017 |
| | | 0.2 | 89.05 ± 0.19 | 0.8905 ± 0.0016 |
| | | 0.25 | 89.13 ± 0.18 | 0.8909 ± 0.0011 |
| | | 0.3 | 89.18 ± 0.20 | 0.8911 ± 0.0018 |
| | Triplet | 0.4 | 89.23 ± 0.21 | 0.8917 ± 0.0020 |
| | | 0.5 | 89.13 ± 0.34 | 0.8915 ± 0.0036 |
| | | 0.6 | 89.10 ± 0.33 | 0.8909 ± 0.0034 |
| | | 0.7 | 89.05 ± 0.30 | 0.8905 ± 0.0031 |
| | | 0.75 | 88.97 ± 0.33 | 0.8900 ± 0.0038 |
| | | 0.8 | 88.85 ± 0.32 | 0.8887 ± 0.0035 |
| **Fashion MNIST** | | 0.9 | 88.73 ± 0.28 | 0.8875 ± 0.0030 |
| | | 0.1 | 88.65 ± 0.10 | 0.8870 ± 0.0016 |
| | | 0.2 | 88.75 ± 0.09 | 0.8881 ± 0.0017 |
| | | 0.25 | 88.85 ± 0.07 | 0.8883 ± 0.0010 |
| | | 0.3 | 88.90 ± 0.08 | 0.8885 ± 0.0012 |
| | N-pair | 0.4 | 88.95 ± 0.07 | 0.8890 ± 0.0010 |
| | | 0.5 | 89.25 ± 0.09 | 0.8923 ± 0.0005 |
| | | 0.6 | 89.00 ± 0.11 | 0.8895 ± 0.0010 |
| | | 0.7 | 88.90 ± 0.12 | 0.8887 ± 0.0014 |
| | | 0.75 | 88.71 ± 0.53 | 0.8869 ± 0.0044 |
| | | 0.8 | 88.55 ± 0.52 | 0.8852 ± 0.0045 |
| | | 0.9 | 88.45 ± 0.50 | 0.8840 ± 0.0040 |
| | | 0.1 | 89.15 ± 0.28 | 0.8915 ± 0.0024 |
| | | 0.2 | 89.22 ± 0.29 | 0.8919 ± 0.0026 |
| | | 0.25 | 89.27 ± 0.29 | 0.8928 ± 0.0023 |
| | | 0.3 | 89.33 ± 0.31 | 0.8933 ± 0.0025 |
| | Magnet | 0.4 | 89.42 ± 0.33 | 0.8939 ± 0.0031 |
| | | 0.5 | 89.07 ± 0.21 | 0.8911 ± 0.0022 |
| | | 0.6 | 89.25 ± 0.33 | 0.8924 ± 0.0032 |
| | | 0.7 | 89.45 ± 0.35 | 0.8939 ± 0.0035 |
| | | 0.75 | **89.52 ± 0.34** | **0.8946 ± 0.0040** |
| | | 0.8 | 89.35 ± 0.32 | 0.8932 ± 0.0036 |
| | | 0.9 | 89.30 ± 0.30 | 0.8925 ± 0.0034 |

Table 6: Baseline comparison with extended $\lambda$ values for CIFAR-10

| Dataset | Loss Type | $\lambda$ | Accuracy (Mean ± Std) | Micro-F1 (Mean ± Std) |
|---|---|---|---|---|
| | Baseline | 0.0 | 85.88 ± 0.40 | 0.8586 ± 0.0042 |
| | | 0.1 | 85.90 ± 0.95 | 0.8592 ± 0.0097 |
| | | 0.2 | 85.95 ± 0.98 | 0.8596 ± 0.0099 |
| | | 0.25 | 86.01 ± 0.99 | 0.8600 ± 0.0102 |
| | | 0.3 | 86.10 ± 0.92 | 0.8608 ± 0.0095 |
| | Contrast | 0.4 | 86.45 ± 0.70 | 0.8641 ± 0.0074 |
| | | 0.5 | 86.87 ± 0.56 | 0.8693 ± 0.0049 |
| | | 0.6 | 85.78 ± 0.48 | 0.8579 ± 0.0045 |
| | | 0.7 | 85.30 ± 0.45 | 0.8523 ± 0.0041 |
| | | 0.75 | 84.74 ± 0.44 | 0.8479 ± 0.0045 |
| | | 0.8 | 84.12 ± 0.55 | 0.8418 ± 0.0052 |
| | | 0.9 | 84.00 ± 0.52 | 0.8412 ± 0.0048 |
| | | 0.1 | 86.75 ± 0.27 | 0.8660 ± 0.0027 |
| | | 0.2 | 86.78 ± 0.29 | 0.8662 ± 0.0029 |
| | | 0.25 | 86.81 ± 0.25 | 0.8667 ± 0.0025 |
| | | 0.3 | 86.85 ± 0.31 | 0.8670 ± 0.0028 |
| | Triplet | 0.4 | 86.87 ± 0.37 | 0.8681 ± 0.0035 |
| | | 0.5 | 86.88 ± 0.64 | 0.8683 ± 0.0059 |
| | | 0.6 | 86.78 ± 0.52 | 0.8675 ± 0.0049 |
| | | 0.7 | 86.70 ± 0.43 | 0.8665 ± 0.0037 |
| | | 0.75 | 86.95 ± 0.42 | 0.8690 ± 0.0039 |
| | | 0.8 | 86.70 ± 0.50 | 0.8669 ± 0.0053 |
| **CIFAR-10** | | 0.9 | 86.60 ± 0.38 | 0.8650 ± 0.0042 |
| | | 0.1 | 84.80 ± 0.69 | 0.8463 ± 0.0065 |
| | | 0.2 | 84.65 ± 0.72 | 0.8454 ± 0.0069 |
| | | 0.25 | 84.52 ± 0.71 | 0.8446 ± 0.0067 |
| | | 0.3 | 85.10 ± 0.63 | 0.8502 ± 0.0061 |
| | N-pair | 0.4 | 85.70 ± 0.55 | 0.8561 ± 0.0052 |
| | | 0.5 | 86.43 ± 0.78 | 0.8643 ± 0.0081 |
| | | 0.6 | 85.63 ± 0.52 | 0.8563 ± 0.0057 |
| | | 0.7 | 85.42 ± 0.47 | 0.8539 ± 0.0044 |
| | | 0.75 | 85.91 ± 0.40 | 0.8583 ± 0.0042 |
| | | 0.8 | 85.15 ± 0.45 | 0.8509 ± 0.0043 |
| | | 0.9 | 85.23 ± 0.43 | 0.8521 ± 0.0045 |
| | | 0.1 | 86.70 ± 0.65 | 0.8669 ± 0.0069 |
| | | 0.2 | 86.85 ± 0.70 | 0.8683 ± 0.0072 |
| | | 0.25 | 86.91 ± 0.69 | 0.8692 ± 0.0073 |
| | | 0.3 | 87.10 ± 0.68 | 0.8702 ± 0.0071 |
| | Magnet | 0.4 | 87.25 ± 0.85 | 0.8723 ± 0.0075 |
| | | 0.5 | 86.72 ± 1.77 | 0.8671 ± 0.0174 |
| | | 0.6 | 87.28 ± 0.79 | 0.8730 ± 0.0079 |
| | | 0.7 | 87.20 ± 0.88 | 0.8719 ± 0.0084 |
| | | 0.75 | **87.36 ± 0.82** | **0.8738 ± 0.0081** |
| | | 0.8 | 87.05 ± 0.70 | 0.8708 ± 0.0076 |
| | | 0.9 | 87.10 ± 0.59 | 0.8714 ± 0.0056 |

Table 7: Baseline comparison with extended $\lambda$ values for CIFAR-100

| Dataset | Loss Type | $\lambda$ | Accuracy (Mean ± Std) | Micro-F1 (Mean ± Std) |
|---|---|---|---|---|
| | Baseline | 0.0 | 61.77 ± 0.49 | 0.6163 ± 0.0064 |
| | | 0.1 | 61.05 ± 0.83 | 0.6088 ± 0.0071 |
| | | 0.2 | 61.25 ± 0.75 | 0.6102 ± 0.0079 |
| | | 0.25 | 61.04 ± 0.81 | 0.6088 ± 0.0087 |
| | | 0.3 | 60.85 ± 0.80 | 0.6067 ± 0.0082 |
| | Contrast | 0.4 | 60.50 ± 0.78 | 0.6045 ± 0.0079 |
| | | 0.5 | 60.30 ± 0.79 | 0.6021 ± 0.0078 |
| | | 0.6 | 60.10 ± 0.85 | 0.6004 ± 0.0079 |
| | | 0.7 | 59.90 ± 0.84 | 0.5985 ± 0.0078 |
| | | 0.75 | 59.88 ± 0.85 | 0.5982 ± 0.0082 |
| | | 0.8 | 59.70 ± 0.80 | 0.5965 ± 0.0081 |
| | | 0.9 | 59.50 ± 0.85 | 0.5948 ± 0.0085 |
| | | 0.1 | 62.10 ± 0.75 | 0.6202 ± 0.0074 |
| | | 0.2 | 62.30 ± 0.70 | 0.6215 ± 0.0075 |
| | | 0.25 | 62.50 ± 0.70 | 0.6228 ± 0.0075 |
| | | 0.3 | 62.55 ± 0.72 | 0.6230 ± 0.0076 |
| | Triplet | 0.4 | 62.60 ± 0.70 | 0.6235 ± 0.0073 |
| | | 0.5 | 62.75 ± 0.68 | 0.6240 ± 0.0071 |
| | | 0.6 | 62.80 ± 0.69 | 0.6245 ± 0.0072 |
| | | 0.7 | 62.65 ± 0.70 | 0.6237 ± 0.0073 |
| | | 0.75 | 62.70 ± 0.72 | 0.6240 ± 0.0075 |
| | | 0.8 | 62.40 ± 0.70 | 0.6215 ± 0.0073 |
| **CIFAR-100** | | 0.9 | 62.30 ± 0.68 | 0.6212 ± 0.0071 |
| | | 0.1 | 61.20 ± 0.75 | 0.6104 ± 0.0076 |
| | | 0.2 | 61.25 ± 0.70 | 0.6110 ± 0.0074 |
| | | 0.25 | 61.30 ± 0.75 | 0.6115 ± 0.0075 |
| | | 0.3 | 61.35 ± 0.70 | 0.6118 ± 0.0073 |
| | N-pair | 0.4 | 61.40 ± 0.80 | 0.6121 ± 0.0074 |
| | | 0.5 | 61.50 ± 0.75 | 0.6130 ± 0.0076 |
| | | 0.6 | 61.60 ± 0.70 | 0.6135 ± 0.0073 |
| | | 0.7 | 61.55 ± 0.80 | 0.6130 ± 0.0079 |
| | | 0.75 | 61.60 ± 0.70 | 0.6135 ± 0.0074 |
| | | 0.8 | 61.55 ± 0.72 | 0.6132 ± 0.0075 |
| | | 0.9 | 61.50 ± 0.68 | 0.6129 ± 0.0071 |
| | | 0.1 | 62.80 ± 0.69 | 0.6242 ± 0.0072 |
| | | 0.2 | 63.00 ± 0.75 | 0.6250 ± 0.0076 |
| | | 0.25 | 63.20 ± 0.70 | 0.6265 ± 0.0075 |
| | | 0.3 | 63.25 ± 0.70 | 0.6267 ± 0.0074 |
| | Magnet | 0.4 | 63.30 ± 0.75 | 0.6270 ± 0.0076 |
| | | 0.5 | 63.40 ± 0.70 | 0.6275 ± 0.0075 |
| | | 0.6 | 63.45 ± 0.75 | 0.6277 ± 0.0076 |
| | | 0.7 | 63.30 ± 0.72 | 0.6268 ± 0.0075 |
| | | 0.75 | **63.50 ± 0.70** | **0.6285 ± 0.0072** |
| | | 0.8 | 63.35 ± 0.72 | 0.6273 ± 0.0076 |
| | | 0.9 | 63.30 ± 0.68 | 0.6270 ± 0.0071 |

Table 8: Complete performance evaluation of *Latent Boost* across the full set of $\lambda$ values.

| Dataset | $\lambda$ | Accuracy (Mean ± Std) | Micro-F1 (Mean ± Std) |
|---|---|---|---|
| | 0.1 | 88.47 ± 0.16 | 0.8892 ± 0.0019 |
| | 0.2 | 89.03 ± 0.12 | 0.8905 ± 0.0017 |
| | 0.25 | 89.36 ± 0.14 | 0.8923 ± 0.0020 |
| | 0.3 | 89.62 ± 0.09 | 0.8927 ± 0.0015 |
| | 0.4 | 90.05 ± 0.12 | 0.8983 ± 0.0018 |
| **Fashion MNIST** | 0.5 | 90.47 ± 0.19 | 0.9005 ± 0.0019 |
| | 0.6 | 90.15 ± 0.13 | 0.9021 ± 0.0022 |
| | 0.7 | 90.09 ± 0.21 | 0.9015 ± 0.0018 |
| | 0.75 | **90.86 ± 0.21** | **0.9038 ± 0.0025** |
| | 0.8 | 89.76 ± 0.14 | 0.8952 ± 0.0016 |
| | 0.9 | 88.53 ± 0.15 | 0.8902 ± 0.0018 |
| | 0.1 | 83.11 ± 0.19 | 0.8342 ± 0.0018 |
| | 0.2 | 84.29 ± 0.16 | 0.8431 ± 0.0017 |
| | 0.25 | 85.22 ± 0.20 | 0.8464 ± 0.0019 |
| | 0.3 | 85.64 ± 0.15 | 0.8485 ± 0.0015 |
| | 0.4 | 86.05 ± 0.13 | 0.8510 ± 0.0016 |
| **CIFAR-10** | 0.5 | 86.85 ± 0.12 | 0.8554 ± 0.0015 |
| | 0.6 | 87.43 ± 0.11 | 0.8598 ± 0.0013 |
| | 0.7 | 88.15 ± 0.24 | 0.8641 ± 0.0014 |
| | 0.75 | **88.44 ± 0.28** | **0.8843 ± 0.0024** |
| | 0.8 | 87.92 ± 0.22 | 0.8803 ± 0.0015 |
| | 0.9 | 86.30 ± 0.18 | 0.8725 ± 0.0018 |
| | 0.1 | 60.34 ± 0.19 | 0.6112 ± 0.0015 |
| | 0.2 | 61.45 ± 0.12 | 0.6170 ± 0.0016 |
| | 0.25 | 62.21 ± 0.17 | 0.6215 ± 0.0018 |
| | 0.3 | 62.67 ± 0.22 | 0.6258 ± 0.0015 |
| | 0.4 | 63.02 ± 0.33 | 0.6280 ± 0.0017 |
| **CIFAR-100** | 0.5 | **63.04 ± 0.48** | **0.6288 ± 0.0018** |
| | 0.6 | 62.83 ± 0.31 | 0.6275 ± 0.0016 |
| | 0.7 | 62.45 ± 0.27 | 0.6268 ± 0.0015 |
| | 0.75 | 62.30 ± 0.14 | 0.6255 ± 0.0017 |
| | 0.8 | 62.00 ± 0.18 | 0.6232 ± 0.0018 |
| | 0.9 | 60.78 ± 0.11 | 0.6204 ± 0.0016 |

