# OpenReview forum: "Latent Boost: Leveraging Latent Space Distance Metrics to Augment Classification Performance"
_ICLR.cc/2025/Conference — ICLR 2025 Conference Withdrawn Submission_

### Official Review · Reviewer_5zW4 · 2024-10-22

**Soundness:** 3
**Presentation:** 3
**Contribution:** 2
**Rating:** 3
**Confidence:** 5

**Summary:**

This paper studies boosting classification performance via leveraging latent space distance metrics. Traditional data-driven classification training focuses on optimizing classification scores in a black-box manner. The authors argue that the internal structural information can significantly elevate training process, but is neglected in conventional probalistic approaches. They introduce Latent Boost, an approach that enhances the cluster separation in the latent space. Results show that Latent Boost not only improves classification metrics but also brings additional benefits of improved interpretatbility with higher silhouette scores and steady-fast covergence.

**Strengths:**

- This paper proposes to enhance cluster separation through optimizing latent space distance metrics during classification training. The idea generally sounds reasonable.
- Latent Boost is simple. Results show that it can enhance performance and interpretability while shrinking computation demand.
- The paper compares Latent Boost with a few other clustering losses. The former shows better performances on three datasets.
- Analyses with different $\lambda$, TSNE plots and Solhouette score are relatively well-presented.

**Weaknesses:**

- Although the authors argue that 'Latent Boost is the first method to integrate distance metrics into the classification loss function', I do feel that it is unexpected that such clustering losses help as they have been widely validated in semi-supervised learning area.

- Latent Boost is based on Magnet loss. A few modifications are designed, such as PCA dimensionality reduction, individual variance, dynamic update rules. However, the authors fail to validate the effectiveness of each component.

- The three datasets experimented in the paper are small. It is unclear whether the proposed loss is effective when there are many training data, such as ImageNet, especially considering the full-supervised learning nature.

- From the paper, the latent representations are extracted from only one layer. While it is reasonable to use the layer before classification head, knowing how to apply distance metrics on early layers seem to be more interesting.

**Questions:**

- In Eq.(9), will $\beta$ become negative when $E>0.2E_{total}$? Figure 3 in Appendix C does not clearly show the situation when Epoch is larger than 100.

- In Eq.(10), should $\epsilon$ be placed in the logarithm or denominator?

- In Ln 377, should it be 'As shown in Section 4.2'?

- For Ln 392-399, could the authors plot out curves of number of principle components throughout training? The authors are suggested to validate the necessity of PCA, e.g., any evidences that (Ln 197) 'it might introduce noise' and the influences of different threshold $T$.

-  In Ln 522, could the authors clarify 'the model is not directly trained on the latent representation data'? I do feel that the loss is directly applied on the latent feature representations to optimize the clustering structure, which is aligned with Silhouette Score.

- What are the number of clusters and how to calcuate cluster means/cluster assignments?

---

### Official Review · Reviewer_eW1a · 2024-10-29

**Soundness:** 2
**Presentation:** 2
**Contribution:** 2
**Rating:** 5
**Confidence:** 3

**Summary:**

The paper introduces Latent Boost, a novel approach that enhances classification performance by leveraging latent space distance metrics. It optimizes models not only for classification metrics but also for sharp class cluster formation within the latent representation, leading to improved F1-Scores and interpretability with minimal additional cost.

**Strengths:**

1). This paper proposes a distance-based loss Latent Boost which is inspired by the Magnet loss, addressing previously overlooked nuisances with dynamic adaptation and discriminative information density.
2). Latent Boost demonstrates a consistent improvement in classification metrics such as F1-Scores across different datasets and model architectures.
3). Latent Boost is able to achieve these performance benefits with minimal additional computing costs and without requiring any data-specific tweaks.

**Weaknesses:**

1).In this paper, three datasets, namely Fashion MNIST, CIFAR-10 and CIFAR-100, are mainly tested. For other more complex datasets, the generalization ability and effect of Latent Boost are not verified in this paper.
2).While Latent Boost is designed to reduce training cycles and increase computational efficiency, the additional computational burden it introduces, such as the amount of computation that can be added when calculating potential spatial distance measures, is not discussed in detail.
3).This paper mainly compares with several traditional distance measurement learning methods, but does not compare with the latest classification methods, which limits the comprehensive evaluation of Latent Boost performance.

**Questions:**

See the above

---

### Official Review · Reviewer_Ufoa · 2024-11-03

**Soundness:** 2
**Presentation:** 3
**Contribution:** 2
**Rating:** 3
**Confidence:** 5

**Summary:**

This paper proposes the Latent Boost method, which aims to enhance classification performance by incorporating distance metric learning in the latent space. The method uses Principal Component Analysis (PCA) for dimensionality reduction before calculating the loss, attempting to reduce noise in early training stages. Additionally, Latent Boost introduces dynamic variance calculation and adaptive hyperparameters for intra-cluster compactness and inter-cluster separation, respectively. Experimental results show some performance improvements over existing methods, highlighting the potential of structured latent representations for classification tasks.

**Strengths:**

1 The method combines PCA with Magnet loss and introduces dynamic variance and adaptive hyperparameters.

2 The paper introduces the concept of dynamically adjusting intra-cluster and inter-cluster factors during training, which is relatively uncommon in metric learning literature.

**Weaknesses:**

1 The core idea—combining PCA with Magnet loss and adding dynamic parameters—lacks substantial novelty. PCA is a basic technique, and the dynamic adjustments are incremental improvements rather than groundbreaking changes. This limits the theoretical contribution of the paper.

2 The dynamic adjustments for intra-cluster and inter-cluster separation might result in overfitting to specific data characteristics. This could negatively impact the model's generalization ability, especially on datasets that significantly differ from those used in the experiments. The paper does not discuss this risk or explore methods to mitigate it.

3 The limited performance gains further suggest that the method lacks the innovation needed to advance the field meaningfully and may merely represent incremental changes to established methods.

**Questions:**

1 How sensitive is the method’s performance to the initial values of α and β? Is extensive tuning required for different datasets, and if so, would this limit the method’s usability?

2 Given the dynamic intra-cluster and inter-cluster adjustments, has the potential for overfitting been explored? Would there be a way to regularize these adjustments to avoid excessive tailoring to specific data characteristics?

---

### Official Review · Reviewer_2nSy · 2024-11-04

**Soundness:** 2
**Presentation:** 3
**Contribution:** 2
**Rating:** 5
**Confidence:** 3

**Summary:**

This paper introduces Latent Boost which levearges latent representation distance metrics to improve the classification training.

**Strengths:**

1. The paper is clearly structured and easy to read.

2. Figure 1 well presents the method pipeline.

**Weaknesses:**

1. No explanantion on the choice of model architecture for three experiments. Sometimes, VGG is used while sometimes ResNet-50 is deployed.

2. All datasets considered are small scale. It's suggested to use more complex datasets.

3. The novelty is somehow limited, by modifying existing magnet loss with PCA. However, the motivation of using PCA is not clearly stated nor ablated using experimental results to show the necessity.

4. The importance of inter- and intra- variance is controled by $\alpha$ and $\beta$. However, they follow two distinct update rule (linearly and exponentially), which is not clearly explained.

5. How is $\lambda$ selected? Table 1 shows the $\lambda$ ranging from [0.5, 0.9] are potentially good, but there is no consistently bset performant $\lambda$. Results on selecting $\lambda$ on the validation set are not presented.

6. All results are conducted in ID test sets. It would be beneficial to show performance gain on OOD test sets.

**Questions:**

See weakness.

---

### Note · Authors · 2024-11-18

I have read and agree with the venue's withdrawal policy on behalf of myself and my co-authors.